



# Dissolved organic matter composition regulates microbial degradation and carbon dioxide production in pristine subarctic rivers

Taija Saarela[1], Xudan Zhu[2], Helena Jäntti[1], Mizue Ohashi[3], Jun'ichiro Ide[4], Henri Siljanen[1], Aake Pesonen[1], Heidi Aaltonen[5], Anne Ojala[6], Hiroshi Nishimura[7], Timo Kekäläinen[8], Janne Jänis[8], Frank Berninger[2], Jukka Pumpanen[1]

[1]Department of Environmental and Biological Sciences, University of Eastern Finland, Kuopio, FI-70210, Finland
[2]Department of Environmental and Biological Sciences, University of Eastern Finland, Joensuu, FI-80101, Finland
[3]School of Human Science and Environment, University of Hyogo, Hyogo, 670-0092, Japan
[4]Department of Applied Chemistry and Bioscience, Chitose Institute of Science and Technology, Chitose, 066-8655, Japan
[5]Department of Agricultural Sciences, University of Helsinki, Helsinki, FI-00014, Finland
[6]Natural Resources Institute Finland (Luke), Helsinki, FI-00790, Finland
[7]Research Institute for Sustainable Humanosphere, Kyoto University, Kyoto, 606-8501, Japan
[8]Department of Chemistry, University of Eastern Finland, Joensuu, FI-80101, Finland

*Correspondence to*: Taija Saarela (taija.saarela@uef.fi)

**Abstract.** Dissolved organic matter (DOM) degradation in freshwater rivers and streams plays a major role in the global carbon cycle. However, little is known about how the source and composition of riverine DOM contribute to the production of greenhouse gases, especially in high-latitude areas with a large proportion of carbon-rich peatlands. Here, we conducted for the first time the combination of molecular-level characterization of terrestrially derived DOM and the potential carbon dioxide ($CO_2$) production measurements in pristine subarctic rivers of Finnish Lapland. 21-day incubation studies were conducted with water samples taken from two rivers differing in DOM content during spring and fall 2018. The changes in the DOM concentration and molecular composition, as well as the $CO_2$ production, were measured. The DOM molecular characterization was carried out using Fourier transform ion cyclotron resonance mass spectrometry (FT-ICR MS). Our results demonstrate efficient mineralization of dissolved organic carbon (DOC) into $CO_2$ in mineral soil associated clearwater river during the incubation, while significantly lower $CO_2$ production per DOC was observed in the brown-water river surrounded by peatlands. The limited degradability in the brown-water river was caused by a large number of terrestrial and aromatic compounds (i.e., highly unsaturated and phenolic compounds, condensed aromatics, and polyphenolics) from surrounding peatlands. In the clearwater river, the percentage of formulas assigned to aliphatics decreased over the incubation, indicating microbial utilization of biolabile DOM. This study highlights the importance of energy-rich, biolabile molecular compounds and the contribution of clearwater systems in the DOM degradation dynamics of subarctic catchments.



## 1 Introduction

Dissolved organic matter (DOM) degradation in freshwater rivers and streams plays a central role in the global carbon (C) cycle
(Cole et al., 2007). The flux of dissolved organic carbon (DOC) and nitrogen (DON) derived from soils and plants is a major link
between terrestrial and aquatic ecosystems (Jaffé et al., 2008). Under the changing climate, the aquatic systems are exposed to
increasing terrestrial organic matter load due to changes in precipitation and air temperature as well as due to reduced sulphur (S)
deposition (Sarkkola et al., 2009; Couture et al., 2011; Pumpanen et al., 2014; Finstad et al., 2016). These increases in terrestrially
derived DOM have the potential to stimulate OC processing and $CO_2$ emissions in freshwater ecosystems across northern
landscapes (Lapierre et al., 2013; Berggren and del Giorgio, 2015). Similarly to lakes that tend to be supersaturated with dissolved
C gases due to terrestrial inputs (Cole et al., 2007), recent studies on riverine gas exchange have indicated also rivers and streams
to be net sources of carbon dioxide ($CO_2$) and methane ($CH_4$) to the atmosphere (Aufdenkampe et al., 2011; Butman and
Raymond, 2011; Huotari et al., 2013; Rocher-Ros et al., 2019).

Over the past decades, the seasonal and annual fluxes of terrestrial DOM have been extensively studied in boreal catchments (e.g.,
Ågren et al., 2007, 2008; Sarkkola et al., 2009; Pumpanen et al., 2014; Rasilo et al., 2015), while studies of DOM fluxes in
northern high-latitude streams are more limited (Wickland et al., 2012; Olefeldt et al., 2013; Giesler et al., 2014; Mzobe et al.,
2018, 2020). So far, molecular composition as a controller of DOM biodegradability has gained little attention in these ecosystems
(e.g., Kellerman et al., 2014; Mostovaya et al., 2017; Hawkes et al., 2018). Previous studies have reported DOM degradation in
freshwaters to be dependent on environmental factors (e.g., temperature, nutrient and oxygen availability, photochemical reactions
and bacterial community composition) and the intrinsic chemical properties, i.e., the DOM composition (Volk et al., 1997;
Tranvik et al., 2001; Bastviken et al., 2004; Vähätalo and Wetzel 2008; Koehler et al., 2012; Smith et al., 2018; Catalán et al.,
2020). The composition of DOM in turn depends on its sources and the transformation of plant compounds into humic-like
substances (Nebbioso and Piccolo, 2012; Kellerman et al., 2014). DOM comprises a wide range of dissolved organic molecules,
being among the most complex molecular mixtures known (Zark and Dittmar 2018). Recent advances in DOM characterization
using Fourier transform ion cyclotron resonance mass spectrometry (FT-ICR MS) have enabled measurements of DOM molecular
composition by identifying thousands of individual molecular formulae present in DOM (Hockaday et al., 2009). FT-ICR MS has
been used for identifying the molecular composition of DOM sources in a variety of aquatic environments, including rivers
(Rogers et al., 2021; Behnke et al., 2022), lakes (Kellerman et al., 2014; Mostovaya et al., 2017) and groundwater (McDonough et
al., 2022). In aquatic ecosystems, the sources of DOM are typically classified into allochthonous (terrestrially derived) and
autochthonous (algal- and macrophyte-derived) sources (Stedmon et al., 2007). Autochthonous DOM has been associated with
lower molecular weight and higher bioavailability compared to allochthonous DOM (Chen and Wangersky, 1996; Henderson et
al., 2008).

Hydrological conditions and catchment characteristics such as soil type, vegetation, and land use can largely affect DOM
biogeochemistry in headwaters (Ågren et al., 2007; Jaffé et al., 2008; Spencer et al., 2008; Kothawala et al., 2015; Catalán et al.,
2016). In subarctic ecosystems, catchment-scale C export and its chemical composition are closely connected to the area covered



by peatlands and the contribution of groundwater (Olefeldt et al., 2013). Peatland-derived DOC in freshwaters tends to have higher concentrations and aromaticity (i.e., lower bioavailability) compared to DOC derived from other terrestrial ecosystems (Ågren et al., 2008; Köhler et al., 2008; Olefeldt et al., 2013). As northern peatlands and forest soils constitute a major C reservoir (Gorham, 1991; Tarnocai et al., 2009), the mobilization of this large C pool due to climate change is expected to stimulate DOC
processing and $CO_2$ emissions in northern aquatic ecosystems (Lapierre et al., 2013).

Here, we aimed to determine how the quantity and composition of DOM influence its microbial degradability in subarctic rivers. More specifically, we aimed 1) to gain a more comprehensive understanding on the composition of riverine DOM at a molecular level, 2) to determine how DOM microbial degradability and relative abundance of bacteria associated with particulate organic matter (POM) differ between brown-water and clearwater river, and 3) to investigate how these factors regulate the potential $CO_2$
production in the water. We hypothesized that DOM in the brown-water river has lower degradability than DOM in the clearwater river because the water originated from peatlands contains recalcitrant humic-like substances. We also hypothesized DOM decomposition to be strongly dependent on terrestrial plant-derived compounds during spring, while the role of aquatic macrophyte- and algae-derived DOM was expected to increase during fall.

## 2 Materials and methods

### 2.1 Site description

The sampling was conducted in a subarctic coniferous forest located in Värriö Strict Nature Reserve (67°44′16″N, 29°38′58″E) in Finnish Lapland close to Värriö Subarctic Research Station (University of Helsinki). The dominating tree species in the area are Scots pine (*Pinus sylvestris* L.), Norway spruce (*Picea abies* ssp. *obovata*) and downy birch (*Betula pubescens* ssp. *pubescens*).
The soils in the area are haplic podzols (FAO 1990) with sand tills. The climate is subcontinental with no underlying permafrost. The mean annual precipitation and air temperature in the study area is 592 mm (Korhonen and Haavanlammi, 2012) and -1 °C (Susiluoto et al., 2008), respectively. The length of the growing season is 105−120 days, and the length of the snow cover period is 200−225 days per year (Pohjonen et al., 2008).

### 2.2 Experimental design

Water sampling was conducted on two sampling occasions (June and October 2018) to investigate the effect of the molecular composition of riverine DOM on its microbial degradability with 21-day incubation studies. Water samples were collected from two rivers that represent contrasting types of catchment characteristics (e.g., vegetation and soil type). Both rivers drain to Barents Sea. The brown-water river (Yli-Nuortti, Fig. S1b) is surrounded by pristine open mires with ~20% catchment peatland coverage.
The ground vegetation in the brown-water river catchment consists of dwarf shrubs (*Betula nana* L., *Salix glauca* L.), flowering



plants (*Geranium sylvaticum* L.), graminoids, and mosses (*Sphagnum* L.). The clearwater river (Kotkakurunoja, Fig. S1c) flows from a steep gorge and is surrounded by mineral soils (<1% catchment peatland coverage). In the clearwater river catchment, the ground vegetation consists of dwarf shrubs (e.g. *Vaccinium myrtillus* L., *Vaccinium vitis idaea* L. and *Empetrum nigrum* L.), lichens (*Cladina* (Nyl.)) and mosses (e.g. *Polytrichum* sp. and *Pleurozium schreberi* (Brid.) Mitt.). Water discharge was

determined based on the continuous water depth measurements carried out by pressure sensors measuring the hydrostatic pressure (Levelogger, Solinst, Georgetown, Canada) in the bottom of the river which was corrected by barometric pressure measurements (Barologger, Solinst, Georgetown, Canada). The water depth measurements were converted to flow rates using channel cross-section, water depth and manual flow rate measurements (Flow Tracker Handheld ADV, SonTek, CA, USA) carried out at sampling locations. The discharge ranged between 0.99–1.01 $m^3$ $s^{-1}$ in the brown-water river and 0.10–0.12 $m^3$ $s^{-1}$ in the

clearwater river during June-October 2018 (Fig. S2).

Two liters of water were collected from six locations of the brown-water river and five locations of the clearwater river (Fig. S1) to pre-combusted (450 °C, 3 h) brown glass bottles that had been washed with 0.01 M nitric acid ($HNO_3$) and rinsed with acetone ($C_3H_6O$) and ultra-pure water (Milli-Q®). After water sampling, surface sediment from one sampling location in both rivers was collected for the inoculum of the incubation experiment.

Immediately after arriving from the field to the research station, river water was filtered through the filtration assembly with pre-combusted (450 °C, 3 h) glass microfiber filters with a nominal pore size of 0.7 µm (Whatman GF/F Glass Microfiber Filters, GE Healthcare Bio-Sciences, Marlborough, MA, USA). The samples were stored at ~0 °C by submerging them in a stream near the research station until further processing.

In the laboratory, surface sediment samples from both rivers were centrifuged gently to separate the particulate matter from the

water, and 10 ml of the separated fluid was then added to 1 l of river water (1:100) as a microbial inoculum to accelerate the microbial activity in the incubation experiment. After that, 300 ml of each sample with the inoculum from the respective river was transferred into a 500 ml glass bottle in three replicates. In addition, one replicate of each water sample without the inoculum and three replicates of Milli-Q water mixed with the inoculum from each river, as well as three replicates of Milli-Q water as a control, were included in the incubation (Fig. S3). Replicates with the inoculum were used to measure the potential production of $CO_2$.

Because no significant difference in the potential $CO_2$ production was observed between the replicates with and without the inoculum, replicates without the inoculum were used to determine the molecular composition of riverine DOM to avoid the possible disturbance resulted from the inoculum on the interpretation of results.

In the beginning of the incubation, 60 ml of water from each incubation bottle was taken to measure the concentrations of DOC and total nitrogen (TN), as well as wavelength specific UV-absorbance at 254 nm ($SUVA_{254}$). In addition, 60 ml of water from the

bottles without the inoculum was taken to measure the DOM molecular composition with FT-ICR MS. After that, the replicates without the inoculum were covered with loose aluminum foil on top of the bottle and stored at +10 °C in the dark for 21 days. The replicates with the inoculum were transferred outside to aerate the samples with ambient air for 15 minutes. Close to each incubation bottle, 25 ml of ambient air was taken for a background sample using 60 ml BD Plastipak™ syringes equipped with a BD Connecta 3-way stopcock valve (Becton, Dickinson and Company, NJ, USA), and the sample was injected with a hypodermic





needle to airtight pre-evacuated 12 ml Exetainer® vials (Labco Ltd., Lampeter, Ceredigion, UK). Immediately after that, the incubation bottles were closed with Butyl Stopper and Aluminum screw caps and stored at +10 °C in the dark for 24 hours, after which they were allowed to stabilize at room temperature (+21 °C) and shaken vigorously for 3 minutes. 25 ml of sample from the gas phase was then taken through Butyl Stopper via a syringe and a needle and injected into a pre-evacuated Exetainer. Thereafter, the bottles were opened, and 60 ml of water was taken to measure SUVA$_{254}$ and the concentrations of DOC and TN in the water.

After the first 24 hours of the experiment, the same procedure was repeated three times (2, 6 and 20 days from the beginning of the experiment), as described in Fig. S3. Between the sampling days, the samples were covered with loose aluminum foil on top of the bottle and stored at +10 °C in the dark. The bottles were carefully inverted for aeration every 2-3 days between samplings.

## 2.3 DOC and TN analyses

The samples for DOC and TN analyses were stored frozen (-18 °C) until further analysis. The concentrations of DOC and TN
were determined with a standard method (SFS-EN 1484) using Shimadzu TOC-V$_{CPH}$ (Shimadzu Corp., Kyoto, Japan). The biodegradable fraction of DOC (% BDOC) was estimated by calculating the change in DOC concentration between the end of incubation (21 d) and the average initial concentration among replicates (0 d) (Catalán et al., 2020).

## 2.4 SUVA$_{254}$

Absorbance measurements were conducted at 254 nm using a 0.01 m quartz cuvette with Shimadzu UV-2401 (Shimadzu Co.,
Kyoto, Japan). Wavelength specific UV-absorbance at 254 nm (SUVA$_{254}$) was calculated as the absorbance divided by DOC concentration, which reflects the aromaticity of the DOM and is inversely related to its biodegradability (Weishaar et al., 2003).

## 2.5 The concentrations of CO$_2$

The CO$_2$ concentrations were measured using Agilent 7890B Gas Chromatograph (Agilent Technologies, Palo Alto, CA, USA) equipped with Gilson liquid handler GX271 autosampler (Gilson Inc., Middleton, WI, USA). The concentrations of CO$_2$ (the gas
phase concentration after 24 hours minus the background ambient concentration) were calculated based on a one-point calibration with standard gas (AGA, Lidingö, Sweden), using Henry's Law and the appropriate temperature relationships (Stumm and Morgan, 1981).

For the calculation of cumulative CO$_2$ production over the 21 days incubation experiment, CO$_2$ production rates ($\mu$mol l$^{-1}$ d$^{-1}$) measured during the four 24 h gas samplings were analyzed. Cumulative sums of CO$_2$ production rates between each consecutive
measurement were calculated for the whole duration of the incubation experiment by linear interpolation. To estimate the CO$_2$ production in relation to DOC content, cumulative sums of CO$_2$ production rates divided by the DOC concentration in the bottle (CO$_2$/DOC ratios) between each consecutive measurement were calculated.



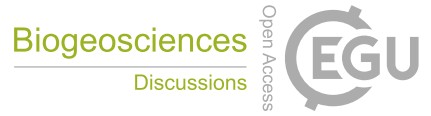

## 2.6 FT-ICR MS analysis

The molecular composition of DOM was analyzed from the samples without the inoculum before and after 21 days incubation
using electrospray ionization (ESI) coupled to ultrahigh-resolution Fourier transform ion cyclotron resonance mass spectrometry
(FT-ICR MS). Samples filtered through glass microfiber filters with a nominal pore size of 0.7 μm (Whatman GF/F Glass
Microfiber Filters) were prepared using the solid phase extraction (SPE) cartridge (Bond Elut® PPL SPE cartridges, Agilent, CA,
USA) to remove inorganic salts (Kim et al., 2003; Dittmar et al., 2008). The samples were diluted with deionized water and
methanol to yield a final sample composition of 50/50 (v/v) of water to methanol. The samples were injected into the FT-ICR MS
(solariX 7.0T, Bruker Daltonics Inc., MA, USA) using a syringe pump, an infusion rate of which was 100 μl h$^{-1}$. All samples were
analyzed in negative ion mode. Ions were accumulated in a hexapole for 0.01 s before they were transferred to the ICR cell, and
the 100 transients collected using a 2 M Word time domain were co-added. All spectra were externally calibrated using the
Tuning Mix standard (Bruker Daltonics Inc., MA, USA) and internally calibrated using the mixture of 10 fatty acids ($C_{15}H_{29}O_2$,
$C_{16}H_{29}O_2$, $C_{16}H_{31}O_2$, $C_{18}H_{35}O_2$, $C_{19}H_{37}O_2$, $C_{20}H_{39}O_2$, $C_{22}H_{43}O_2$, $C_{24}H_{47}O_2$, $C_{26}H_{51}O_2$, $C_{30}H_{59}O_2$). The samples were analyzed three
times per sample, and the peak list of mass-to-charge ratio (m/z) shared among the three analytical replicates was extracted. Mass
lists were produced using a signal-to-noise ratio (S/N) cut-off of 5. Isotope peaks were removed from the list. The molecular
formula calculator (Molecular Formula Calculator ver. 1.0; ©NHMFL, 1998) was used to assign an expected molecular formula
for each m/z value with a mass accuracy ≤ 0.5 ppm. The m/z values in the range of 150-500 were inserted into the molecular
formula calculator. The following conditions were used for formula assignment: C = 1 − ∞; H = 1 − ∞; O = 1 − ∞; N = 0 − 2; S =
0; P = 0; double bond equivalence (DBE) ≥ 0. Since high errors are associated with the assignments containing S and P
(Mostovaya et al., 2017), these formulae were excluded from further analysis. After the formula assignment with the molecular
formula calculator, molecular formulas not likely to be observed in natural water were eliminated based on rules described in
Kujawinski and Behn (2006) and Wozniak et al. (2008).

The modified aromaticity index (AI$_{mod}$), reflecting the degree of aromaticity, was calculated as AI$_{mod}$ = [1 + C − ½O − ½(N +
H)]/(C − ½O − N), where C, H, O, and N refer to a number of respective atoms per molecule. The formula for AI$_{mod}$ was modified
by Mostovaya et al. (2017) based on Koch and Dittmar (2006). Compound classes were assigned using AI$_{mod}$ and oxygen-to-
carbon (O/C) and hydrogen-to-carbon (H/C) ratios as follows: polyphenolics (0.5 < AI$_{mod}$ ≤ 0.66); condensed aromatics (AI$_{mod}$ >
0.66); highly unsaturated and phenolics (HUPs; AI$_{mod}$ ≤ 0.5, H/C < 1.5, O/C ≤ 0.9); aliphatic (1.5 ≤ H/C ≤ 2.0, O/C ≤ 0.9 and N =
0); sugar-like (O/C > 0.9); and peptide-like compounds (1.5 ≤ H/C ≤ 2.0, and N > 0).


## 2.7 16S qPCR analysis

The number of bacteria in the water samples was analyzed by using 16S qPCR. The glass microfiber filters with a nominal pore
size of 0.7 μm (Whatman GF/F Glass Microfiber Filters) used in the filtration of river water samples (n=6 in the brown-water
river and n=5 in the clearwater river) were stored frozen at −18 °C until further treatment. The filters were transferred to



BeadBeater tubes with a sterilized spoon and homogenized with BeadBeater (BioSpec Products Inc., Bartlesville, OK, USA). To store the sample material, BeadBeater lysis buffer was added to the tubes. Clean Whatman GF/F Glass Microfiber Filters were used as a control and treated similarly with samples. For DNA extraction, homogenized filters were transferred into a pre-cooled Lysing tube E (MP Biomedicals, USA) with a sterilized spoon. For a detailed description of DNA extraction protocol, see Siljanen et al. (2019). The 16S rRNA gene in water DNA extracts was PCR-amplified using F338-forward and R518-reverse primers.

Reactions were carried out using 16S qPCR X1 Mastermix (Table S2). For detailed steps in the 16S rRNA protocol, see Table S3. It has to be acknowledged that the number of bacteria used to compare the relative abundances between the water samples is an approximation based on particulate organic matter (POM) because this analysis did not include small microbes due to the use of the glass microfiber filter with a 0.7 μm nominal pore size.

**2.8 Statistical analyses**

To test for differences between rivers and sampling occasions in the cumulative $CO_2$ production, $CO_2/DOC$ ratios, DOC and TN concentrations and $SUVA_{254}$ (0 and 21 days), we applied a two-way ANOVA, coupled with Tukey's HSD post-hoc test (*aov* and *TukeyHSD* functions in R). Furthermore, to test for differences between rivers and sampling occasions in the DOM molecular composition (i.e., m/z ratio, relative peak intensity, H/C and O/C, and $AI_{mod}$ values of detected molecular formulas before and after the incubation), a two-way ANOVA coupled with Tukey's HSD post-hoc test was applied. In all cases, variables were tested

for normality using a Shapiro–Wilk test (*shapiro.test* function in R). For variables failing the normality test ($SUVA_{254}$, the number of 16S rRNA gene copies, m/z ratio, relative peak intensity, H/C and O/C, and $AI_{mod}$ values of detected molecular formulas), we applied a Kruskal-Wallis test followed by a Wilcoxon's test (*kruskal.test* and *pairwise.wilcox.test* functions in R). Spearman rank correlations were used to assess the relationship between the cumulative $CO_2$ production, $CO_2/DOC$ ratios, BDOC (%), DOC and TN concentrations, $SUVA_{254}$, and the percentages of van Krevelen diagram-derived classification groups (*cor()* function in R). All

statistical analyses were performed using R (version 3.6.2; R Core Team 2020) in RStudio (RStudio Team 2020).

**3 Results**

**3.1 Microbial degradability of DOM**

The DOC concentration was significantly higher in the brown-water river than in the clearwater river after 21 days of incubation

($p < 0.001$), while at the beginning of incubation, the differences in the DOC concentration were smaller than the variability (Fig. 1a, b). In both rivers, the concentration of DOC decreased over the incubation, but the decrease in DOC was statistically significant only in the clearwater river during October ($p = 0.004$; Fig 1b). The concentration of TN was significantly higher in the brown-water river than in the clearwater river ($p = 0.04$ at the beginning of incubation and $p = 0.01$ after the incubation; Fig. 1c, d). In both rivers, the concentration of TN decreased over the incubation during October ($p < 0.001$ in the brown-water river and p

$= 0.005$ in the clearwater river; Fig 1d), while the decreases in TN were not statistically significant during June (Fig. 1c).





SUVA$_{254}$, reflecting the aromaticity of the DOM, was higher in the brown-water river both before and after the incubation (p = 0.003 and p = 0.002, respectively; Fig. 1e, f).

**Figure 1. The concentrations of DOC (µmol l$^{-1}$) in (a) June and (b) October, TN (µmol l$^{-1}$) in (c) June and (d) October, and SUVA$_{254}$ (mg l$^{-1}$ m$^{-1}$) in (e) June and (f) October over the incubation period (21 days). Values are presented as averages ± standard deviations (n = 6 in the brown-water river and n = 5 in the clearwater river).**



There were no statistically significant differences in the cumulative $CO_2$ production between the clearwater and brown-water
rivers. In the clearwater river, the cumulative $CO_2$ production decreased from June to October ($p = 0.045$; Fig. 2a, b), while the
cumulative production of $CO_2$ was not significantly different between June and October water samples of the brown-water river
(Fig. 2a, b). The cumulative ratios of $CO_2/DOC$, reflecting the degradation of DOC into $CO_2$, were higher in the clearwater river
than in the brown-water river ($p < 0.001$; Fig. 2c, d). In both rivers, the cumulative $CO_2/DOC$ ratio decreased from June to
October ($p < 0.001$; Fig. 2c, d). The biodegradable fraction of DOC (%BDOC) was significantly higher in the clearwater river
than in the brown-water river ($p < 0.001$; Fig. 2e, f).







**Figure 2. Cumulative CO₂ production in (a) June and (b) October, cumulative CO₂/DOC ratio (the CO₂ production per DOC) in (c) June and (d) October, and the proportion of BDOC (%) in (e) June and (f) October over the incubation period (21 days). Boxplots show the median (horizontal line), upper and lower quartile, as well as the smallest and largest value (n = 6 in the brown-water river and n = 5 in the clearwater river).**



## 3.2 Molecular composition of DOM

The molecular composition of river water samples was determined on a molecular formula level by FT-ICR MS. The mass-to-
charge (m/z) ratios were significantly higher in the clearwater river than in the brown-water river during June ($p < 0.001$; Table 1),
while the brown-water river had significantly higher m/z ratios in October ($p < 0.001$). The m/z ratios generally decreased over the
incubation ($p < 0.001$). However, in the clearwater river, there was an increase in the m/z ratios over the incubation during
October ($p < 0.001$). While no significant differences in O/C ratios were detected between the rivers at the beginning of
incubation, O/C ratios were significantly higher in the brown-water river than in the clearwater river after the incubation ($p < $
$0.001$; Table 1). In the clearwater river, O/C ratios decreased over the incubation ($p = 0.048$ in June and $p = 0.007$ in October).
H/C ratios were significantly higher in the clearwater river both before and after the incubation during October ($p = 0.02$ and $p < $
$0.001$, respectively). The percentage of biolabile high H/C molecular formulas (>1.5) ranged from 22% to 30% in the clearwater
river and from 19% to 22% in the brown-water river (Table 1).

The relative peak intensities of detected molecular formulas were higher in the clearwater river than in the brown-water river
during both spring and fall ($p < 0.001$; Fig. 3). In the brown-water river, the relative peak intensities decreased over the
incubations ($p < 0.001$). In the clearwater river, there was an increase in the relative peak intensities over the incubation during
June ($p <0.001$), while the relative peak intensities decreased over the October incubation ($p < 0.001$). The modified aromaticity
index ($AI_{mod}$) was significantly higher in the brown-water river than in the clearwater river after the October incubation ($p < $
$0.001$; Table 1), whereas in June, there were no statistically significant differences in $AI_{mod}$ between the rivers.


Table 1. Summary of molecular formulas derived from the FT-ICR MS analysis of river water samples before (day 0) and after the
incubation (day 21). Values are presented as averages ± standard deviations (n = 6 in the brown-water river and n = 5 in the clearwater
river).

| Sampling month | Sample | Total assigned formulas | Mean m/z ratio | Mean H/C ratio | Mean O/C ratio | Mean $AI_{mod}$ | H/C > 1.5 (%) |
|---|---|---|---|---|---|---|---|
| **June 2018** | Brown-water river day 0 | 269 ± 38 | 364 ± 77 | 1.20 ± 0.42 | 0.38 ± 0.20 | 0.31 ± 0.31 | 20.1% |
| **June 2018** | Brown-water river day 21 | 332 ± 245 | 356 ± 76 | 1.18 ± 0.39 | 0.39 ± 0.19 | 0.33 ± 0.28 | 22.3% |
| **June 2018** | Clearwater river day 0 | 246 ± 40 | 377 ± 83 | 1.19 ± 0.40 | 0.38 ± 0.20 | 0.31 ± 0.35 | 22.9% |
| **June 2018** | Clearwater river day 21 | 235 ± 58 | 374 ± 79 | 1.20 ± 0.38 | 0.36 ± 0.20 | 0.31 ± 0.31 | 23.4% |
| **October 2018** | Brown-water river day 0 | 140 ± 67 | 390 ± 78 | 1.19 ± 0.39 | 0.41 ± 0.22 | 0.29 ± 0.32 | 18.6% |
| **October 2018** | Brown-water river day 21 | 187 ± 148 | 364 ± 77 | 1.16 ± 0.40 | 0.42 ± 0.19 | 0.33 ± 0.27 | 20.7% |
| **October 2018** | Clearwater river day 0 | 125 ± 87 | 332 ± 66 | 1.25 ± 0.40 | 0.42 ± 0.16 | 0.28 ± 0.25 | 30.0% |
| **October 2018** | Clearwater river day 21 | 163 ± 55 | 365 ± 75 | 1.26 ± 0.40 | 0.39 ± 0.19 | 0.28 ± 0.26 | 21.7% |




Number of molecular formulas assigned to each compound class varied between the rivers (Fig. 3, 4). In both rivers, DOM composition was dominated by unsaturated and phenolic compounds (HUPs), which accounted for over 50% of assigned molecular formulas (Fig. 4b, d). While the percentage of formulas assigned to aliphatics was similar in both rivers during June (14-15%; Fig. 4b), the clearwater river had a clearly higher percentage of aliphatics in October (24% in the clearwater river and

12% in the brown-water river; Fig. 4d). Furthermore, the percentage of formulas assigned to peptide-like compounds was higher in the clearwater river (9%) than in the brown-water river (4%) during June (Fig. 4b). Both in June and October, the brown-water river had a higher proportion of condensed aromatics (12% in June and 11% in October) and polyphenolics (13% in June and 15% in October) compared to DOM pool in the clearwater river (Fig. 4b, d).

In June, the percentage of formulas assigned to HUPs decreased by 7% in the brown-water river after 21 days of incubation (Fig.

4b), while other compound classes in the DOM pool of brown-water river showed limited changes over the incubation. In October, the percentage of condensed aromatics decreased by 3% and polyphenols by 2% in the brown-water (Fig. 4d). In the clearwater river, the changes in DOM molecular composition were most evident in formulas assigned to aliphatics, which decreased by 2% in June (Fig. 4b) and by 6% in October (Fig. 4d).










**Figure 3. Molecular element ratios (van Krevelen diagrams) from the FT-ICR MS analysis of river water samples in June (a) Brown-water river before incubation, (b) Brown-water river after 21-day incubation, (c) Clearwater river before incubation, and (d) Clearwater river after 21-day incubation, and in October (e) Brown-water river before incubation, (f) Brown-water river after 21-day** incubation, **(g) Clearwater river before incubation, and (h) Clearwater river after 21-day incubation. Different colors represent compound groups assigned using** $AI_{mod}$ **and oxygen-to-carbon (O/C) and hydrogen-to-carbon (H/C) ratios as follows: polyphenolics (0.5 < $AI_{mod}$ ≤ 0.66); condensed aromatics ($AI_{mod}$ > 0.66); highly unsaturated and phenolics (HUPs; $AI_{mod}$ ≤ 0.5, H/C < 1.5, O/C ≤ 0.9); aliphatic (1.5 ≤ H/C ≤ 2.0, O/C ≤ 0.9 and N = 0); sugar-like (O/C > 0.9); and peptide-like compounds (1.5 ≤ H/C ≤ 2.0, and N > 0). Different sizes of bubbles represent peak intensities normalized to the sum of all signal intensities in each sample.**



**Figure 4. Comparison of average values of the number of m/z peaks (± standard error) from the FT-ICR MS analysis in each compound group in (a) June and (c) October, and percentages of molecular formulas assigned to each compound group in (b) June and (d) October. BW day 0 = Brown-water river before incubation, BW day 21 = Brown-water river after 21-day incubation, CW day 0 = Clearwater river before incubation, and CW day 21 = Clearwater river after 21-day incubation. Compound groups were assigned using** $AI_{mod}$ **and oxygen-to-carbon (O/C) and hydrogen-to-carbon (H/C) ratios as follows: polyphenolics (0.5 < $AI_{mod}$ ≤ 0.66); condensed aromatics ($AI_{mod}$ > 0.66); highly unsaturated and phenolics (HUPs; $AI_{mod}$ ≤ 0.5, H/C < 1.5, O/C ≤ 0.9); aliphatic (1.5 ≤ H/C ≤ 2.0, O/C ≤ 0.9 and N = 0); sugar-like (O/C > 0.9); and peptide-like compounds (1.5 ≤ H/C ≤ 2.0, and N > 0).**



## 3.3 Bacterial abundance in rivers

The number of bacterial 16S rRNA gene copies quantified from October water samples ranged from $4.91 \times 10^3$ $l^{-1}$ to $5.78 \times 10^6$ $l^{-1}$
in the clearwater river (Fig. 5). In the brown-water river, the concentration of bacterial 16S rRNA genes varied between $1.74 \times 10^4$
$l^{-1}$ and $6.10 \times 10^5$ $l^{-1}$.

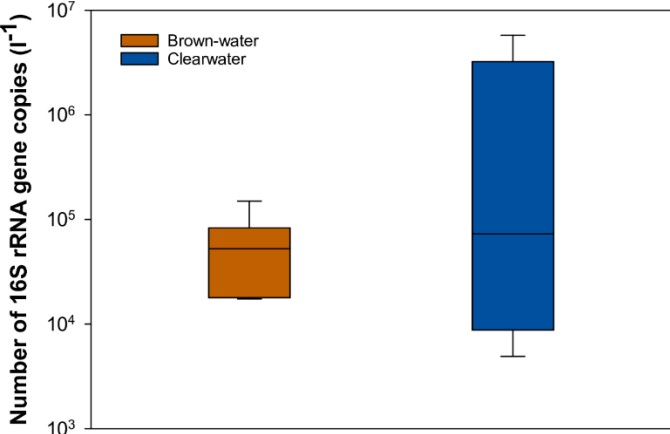

**Figure 5. Abundance of bacterial communities estimated by quantifying the bacterial 16S rRNA gene copies ($l^{-1}$) in brown-water and**
**clearwater river during October 2018. Boxplots show the median (horizontal line), upper and lower quartile, as well as the smallest and**
**largest value (n = 6 in the brown-water river and n = 5 in the clearwater river).**

## 3.4 Relationships between microbial degradation and molecular composition of DOM

The cumulative $CO_2$ production was positively correlated with $SUVA_{254}$ (0.56, p = 0.01 at the beginning and 0.52, p = 0.02 after
the incubation) and the number of molecular formulas assigned to peptide-like compounds (0.42, p = 0.03 at the beginning and
0.49, p = 0.02 after the incubation; Table S4). In addition, the cumulative $CO_2$ production was significantly correlated with the
concentration of TN (0.58, p = 0.006) and the number of condensed aromatics (0.49, p = 0.02) after 21 days of incubation. The
cumulative $CO_2$ production per DOC ($CO_2$/DOC ratio) was positively correlated with %BDOC (0.52, p = 0.02) and the number of
peptide-like compounds (0.55, p = 0.01 before and 0.69, p < 0.001 after 21 days of incubation). Furthermore, the cumulative
$CO_2$/DOC ratio had a negative correlation with the concentration of TN (-0.47, p = 0.03) at the beginning of incubation.

## 4 Discussion

The combination of potential $CO_2$ production measurements and molecular-level characterization of DOM was conducted in two
subarctic river ecosystems that represent contrasting types of catchment characteristics. The results indicate efficient



mineralization of DOC into $CO_2$ in mineral soil-associated clearwater river during 21 days of incubation, while significantly lower $CO_2$ production per DOC was observed in the brown-water river surrounded by peatlands (Fig. 2c, d). The higher microbial degradation of mineral soil-associated DOM was also supported by significantly lower recalcitrance ($SUVA_{254}$) in the clearwater river compared to the brown-water river (Fig. 1e, f). These results suggest the presence of more labile, easily biodegradable compounds in the DOM pool of the clearwater river (Kalbitz et al., 2003; Roth et al., 2019). Indeed, we observed a large number

of biolabile high H/C (>1.5) molecular formulas in the clearwater river (22-30% of all assigned molecular formulas; Table 1), including aliphatic and peptide-like compounds (Fig. 4). These compounds, together with relatively high bacterial abundances of 16S rRNA gene copies (Fig. 5), provide evidence for microbial activity in the clearwater river (D'Andrilli et al., 2015; Spencer et al., 2015; Behnke et al., 2022). The strong positive correlations between the cumulative $CO_2$ production and the number of peptide-like compounds further support this conclusion.

In October, the aliphatic rich signature in the clearwater river might also indicate DOM of algal origin (Chen et al., 2016; Liu et al., 2020). Algal-derived DOM is typically more bioavailable than terrestrial or plant-derived DOM (Guillemette et al., 2013), and it supports a greater bacterial growth efficiency (Kritzberg et al., 2005). The decreases in the percentages of aliphatics over the incubation (Fig. 4b, d) further suggest that as energy-rich and highly reduced compounds, aliphatics were important substrates for aquatic bacterial metabolism in the clearwater river (Chróst and Gajewski, 1995; Berggren et al., 2010; Parrish, 2013; Spencer et

al. 2015; Textor et al., 2019).

In the brown-water river, the lower $CO_2$ production per DOC together with higher $SUVA_{254}$ values elucidate the limited degradability of peatland-derived DOM (Fig. 1e, f; Fig. 2c, d). This observation was further supported by the significantly lower fraction of biodegradable DOC (BDOC%) in the brown-water river compared to the clearwater river (Fig. 2e, f). Furthermore, the limited microbial degradability in the brown-water river is in agreement with a large number of terrestrial and aromatic

compounds (i.e., highly unsaturated and phenolic compounds, condensed aromatics and polyphenolics) that were most likely originated from surrounding peatlands (Fig. 4). The dominance of these compounds in the DOM mixture of brown-water river indicates vascular plant-derived lignin compounds from peatlands (Spencer et al., 2008; Mostovaya et al., 2017; Behnke et al., 2022). The presence of lignins is typical of colored, peatland-derived DOM with high aromatic content (Ågren et al., 2008; Zark and Dittmar, 2018). As phenolic polymers, lignins are traditionally considered to be degraded at a lower rate than other plant-

derived compounds (e.g., cellulosic and non-cellulosic polysaccharides and proteins) (Martin et al., 1980; Haider, 1992).

Both before and after 21 days incubation, the studied rivers were strongly dominated by highly unsaturated and phenolic compounds (HUPs), as is typical of DOM globally (Spencer et al., 2015; Rogers et al., 2021; Behnke et al., 2022). HUPs have been linked to vascular-plant derived lignin degradation products as well as chemically stable carboxylic-rich alicyclic molecules (CRAM; Hertkorn et al., 2006; Roth et al., 2019). Surprisingly, these compounds appear to be more stable over time than

compounds with higher aromaticity (i.e., condensed aromatics and polyphenolics; Kellerman et al., 2014; Mostovaya et al., 2017). Even though peatland-derived DOM was associated with limited degradability in our 21-day incubation study, DOM from different sources can become similarly bioavailable at timescales of months to years (Vähätalo and Wetzel, 2008; Koehler et al., 2012). This is because compared to terrestrial DOM with higher aromaticity, the consumption of more labile fractions of DOM





mainly occurs on shorter timescales due to the rapid turnover of the low-molecular-weight C compounds (Sundh, 1992; Chen and
Wangersky, 1996; Kalbitz et al., 2003). On timescales longer than a month, the compositional similarities between the rivers (e.g.,
the predominance of HUPs) could eventually lead to diminished differences in DOM decomposition rates.

Northern high-latitude rivers exhibit a strongly seasonal discharge with maximum discharge occurring during the spring freshet
(Raymond et al., 2007; Holmes et al., 2008; Spencer et al., 2008). The DOM exported during the spring freshet has been found to
be younger and more labile than during other seasons (Neff et al., 2006; Raymond et al., 2007; Holmes et al., 2008). Indeed, the
results in the studied rivers indicate higher $CO_2$ production per DOC during spring than later in the year (Fig. 2c, d). However, the
results also suggest that DOM exported during spring was associated with higher aromaticity compared to fall (Fig. 3), similar to
previous Arctic river studies showing the chemical paradox of spring freshet DOM being both biolabile and highly aromatic (e.g.,
Spencer et al., 2008; Behnke et al., 2022). This might be related to the dominant source of DOM shifting from surface runoff
during the spring freshet to less aromatic groundwater during low-discharge and baseflow periods (Guo and MacDonald, 2006;
Neff et al., 2006; Spencer et al., 2008). On the other hand, the increase of benthic macrophytes of both rivers during fall, as
observed by ocular perception, might also explain the transition toward less aromatic DOM, because macrophyte-derived DOM is
known to have lower aromaticity and higher proportion of low-molecular-weight molecules than terrestrially derived DOM (Qu et
al., 2013; Liu et al., 2020; They et al., 2012). Furthermore, relatively high DOC concentrations detected during October might
have originated from DOC production by macrophytes in these rivers (Reitsema et al., 2018), because some macrophyte species
can release DOC from their roots to stimulate endomycorrhizal or microbial activity in the sediment (Wigand et al., 1998). The
variations in DOM aromatic content during different seasons thus highlight that the degradation of these complex DOM pools is
not solely controlled by their chemical composition and concentrations but also by environmental conditions and photosynthetic
activity.

Taken together, our data indicates that riverine DOM from subarctic peatlands can be considered relatively stable in terms of in-
stream processing. While climate-induced changes in vegetation cover, plant biomass and water discharge are predicted to
accelerate the delivery of terrestrial plant-derived OC to aquatic systems (Butman et al., 2012; Bragazza et al., 2013), thus
promoting the browning of surface waters (Roulet and Moore, 2006; Finstad et al., 2016; Fork et al., 2020), our results suggest
that high loads of terrestrial plant-derived aromatic compounds will not necessarily stimulate OC processing and aquatic $CO_2$
production in northern landscapes surrounded by peatlands. Low-molecular-weight compounds derived from mineral soils, on the
other hand, can be much more mobile than high-molecular-weight aromatic compounds from peatlands (Kaiser et al., 2002), and
they have a short turnover time in receiving waters, which enables continuous export of dissolved labile metabolites to aquatic
systems (Berggren et al., 2010). The results reported here demonstrate the role of biolabile molecular compounds in the DOM
degradation dynamics of subarctic river waters. This emphasizes the relevance of mineral soil-associated clearwater systems for
DOM processing in northern high-latitude catchments.




## 5 Conclusions

In this study, we conducted for the first time the combination of molecular-level characterization of DOM and the potential $CO_2$ production measurements in two contrasting subarctic river ecosystems located in the pristine areas of Finnish Lapland and draining to Barents Sea. In the clearwater river, the percentage of molecular formulas assigned to aliphatics decreased over the
incubation, simultaneously with efficient mineralization of DOC into $CO_2$. This highlights the importance of biolabile molecular compounds and the contribution of clearwater systems in the DOM degradation dynamics of subarctic catchments. While subarctic peatland catchments are associated with high DOM content, peatland-derived DOM in the brown-water river had lower degradability compared to DOM in the clearwater river. This was caused by a large number of less biodegradable, vascular plant-derived compounds in the DOM mixture. Since this peatland-derived DOM can be considered relatively stable in terms of in-
stream processing, we suggest that increasing loads of terrestrial plant-derived aromatic compounds due to climate change will not necessarily accelerate $CO_2$ production in subarctic freshwaters surrounded by peatlands.

## Data availability

All data used in this study will be freely available at Zenodo (https://zenodo.org/) after the acceptance of the manuscript. The data is also available from the authors on request.

## 420 Author contribution

TS, HJ, JP, FB and AO contributed to the study conception and design. Field work was conducted by TS, JP, FB and AO. Laboratory analyses were performed by TS, XZ, HJ, JI, AP, HS, and HA. TS, JI, MO, FB, TK, and JJ analyzed the data. TS wrote the first draft of the manuscript. All authors read, commented and approved the final manuscript.

## Competing interests

The authors declare that they have no conflict of interest.

## Acknowledgements

This work was supported by Doctoral Programme in Environmental Physics, Health and Biology, University of Eastern Finland; the Academy of Finland [275127, 307331, 326818, 323997]; MEXT KAKENHI [20346689, JP15K16115, JP18K11623, JP19K22444]; Japan Society for the Promotion of Science [JSBP120-209933, FY2018]; Research Institute for Sustainable
Humanosphere (RISH), Kyoto University; and joint funding by Olvi Foundation, Jenny and Antti Wihuri Foundation, and Saastamoinen Foundation. We thank Johanna Kerttula and Inga Paasisalo for help in laboratory analyses.



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
