# Peer review of "Dissolved organic matter composition regulates microbial degradation and carbon dioxide production in pristine subarctic rivers"

_Biogeosciences, 2022_

## Referee Comment (RC2)

General comments:

The manuscript "Dissolved organic matter composition regulates microbial degradation and carbon dioxide production in pristine subarctic rivers" by Saarela et al. provides a nice comparison between DOM and $CO_2$ production in the a clear water vs brown water systems in the high latitude watershed. Studies linking DOM and $CO_2$ has become very crucial in recent years with increasing amount of greenhouse gas emission from inland waters, and this manuscript provides valuable findings on the topic. The manuscript includes a number of advanced techniques including $CO_2$ measurement, FT-ICR-MS, and qPCR, which combined with appropriate statistical analysis seems adequate to support the major findings. The manuscript will be valuable addition in the field of aquatic biogeochemistry and will benefit the readers of Biogeosciences. I have a few suggestions for the authors to consider before the final publication of the manuscript.

Specific comments:

1. In lines 108-109, the author mention about adding an inoculum from the surface sediment. Since river water samples are usually added as inoculum for incubation experiments, please provide a brief explanation for adding inoculum from the sediment.

2. Also, a previous meta-analysis study on BDOC measurement method (Vonk et al. 2015) reported no significant difference between the BDOC measured with or without inoculum when a 0.7 $\mu$m filter like GFF is used. The author could have avoided adding inoculum since enough microbes pass through the filter required for microbial degradation.

3. The incubation experiment for measurement of $CO_2$ is quite interesting. A little more details on the calculations methods or showing the actual data on a SI table would be helpful for the readers. Also, why did the author use a one-point calibration when at least two point is more usual for calibration.

4. Please add a relevant reference for the compound classes assignment criteria (line 181-184).

5. In the results, I see a lot of statistical analysis results (i.e., p values); however, I miss seeing the actual values of the major parameters particularly in comparison between seasons or water types. Including some actual values for DOC, $CO_2$ etc. in the result section would improve the readability of the manuscript, whereas adding the statistical results in the figure would also be helpful.

6. The discussion is well supported with references, but you may add the following reference to support the findings on lability of molecular composition (example: line 344-349). Begum et al. 2022 (https://doi.org/10.1016/j.watres.2022.119362).

---

## Author Comment (AC1)

**Manuscript No.: bg-2022-225**

**Response to RC1**

Dear Reviewer #1,

We thank the reviewer for the insightful comments and suggestions for this manuscript. We have incorporated most of the suggestions made by the reviewer. Please see below for a point-by-point response to the reviewer's comments.

**Reviewer's comments to the authors:**

**Reviewer #1**

**In "Dissolved organic matter composition regulates microbial degradation and carbon dioxide production in pristine subarctic rivers," Saarela et al. examine differences in DOM composition and CO2 production between clearwater and brownwater rivers in subarctic Finish Lapland. They find clearwater river DOM more biolabile, with interesting implications for how future increases in terrestrial DOM inputs may not increase CO2 fluxes. The manuscript covers an understudied region and could be a useful contribution to our processed based understanding of CO2 production. However, there are a few major issues with the study design and methods that will need addressing and clarifying in the manuscript itself.**

Author response: We thank the reviewer for the feedback and for pointing out the issues concerning the study design and methodology of this study.

**Incubation design: As laid out in figure S3, the overall incubation design causes me some concern and I do not think its potential implications are addressed in the paper. While I understand the desire to use the incubation (C in the Figure S3) without added inoculum for FT-ICR MS so as to avoid the influence of the inoculum on molecular formulae, I am worried about the direct comparison between CO2 analyses conducted in**

**a set of bottles with an inoculum added and ICR metrics for a final time point derived from a completely different set of bottles with no inoculum added. To me it seems very much like comparing apples and oranges. I understand that in lines 120-122, the authors explain that there was no difference in potential CO2 production between bottles with the inoculum and those without, though I admit I do not understand how this was analyzed since Figure S3 shows that treatment 2 (C, the one with no inoculum) never had its CO2 measured, so I am not sure where that potential CO2 production measurement comes from. Please explain this in the manuscript. But CO2 production is not the only change that the presence of microbes can cause—they can change the composition of DOM through partial consumption, and it is possible the different size classes of microbes present in the filtered (treatment 2) and inoculum (treatment 1) bottles would do different things to DOM composition.**

**Thus, I do not think it is valid to compare CO2 production to ICR composition at the end of the incubation period, after the processing both sets of bottles have undergone has diverged. Comparing it to the composition at the beginning of the experiment makes sense, since those were the molecular starting conditions that led to the CO2 production. It would greatly strengthen the manuscript if you removed reference to the final timepoint for FT-ICR MS (or at least the comparison of that time point to CO2 production), and focused on the relationship between the starting FT-ICR MS data.**

Author response: We thank the reviewer for pointing this out. We understand the reviewer's concerns about the direct comparison between the incubation bottles with and without an inoculum added. In addition, we agree that the study design needed more clarification. For comparison of the $CO_2$ production in bottles with and without inoculum, the $CO_2$ production was measured from bottles without inoculum at the beginning of incubation and at the final time point (21 days). We have clarified this in the manuscript (page 5 lines 122-138) and updated Figure S3 concerning different treatments and their measurements (page 5 in the supplementary material).

We also thank the reviewer for the suggestions concerning the direct comparison of $CO_2$ production to DOM compositional changes during the incubation (i.e., the FT-ICR MS data after the incubation). In the revised manuscript, we have focused on discussing the molecular "starting conditions" (incubation day 0) and their impact on the microbial biodegradability and potential $CO_2$ production in these rivers. Furthermore, in chapter 3.4 (Relationships between

microbial degradation and molecular composition of DOM), we have excluded the final time point (21 days) of FT-ICR MS results and focused on presenting the correlations between the $CO_2$ production and the molecular compound groups at the beginning of incubation (page 5 lines 336-343 and Table S4 in the supplementary material).

**Contaminant peaks: The extremely high relative abundance peaks in the top left corner of all your van Krevlen diagrams that you attribute to the aliphatics class look to me like classic surfactant peaks that are often added to FT-ICR MS analysis through the SPE process. They are often of the O3S1 class, and are several series of homologous peaks (separated by a CH2 unit). If they are these common contaminants, they should be removed from the analysis. This will change the % relative abundances of your compound classes, since these currently (and inaccurately, I believe) dominate.**

Author response: We thank the reviewer for this comment. We carefully checked each molecular formula with extremely high relative peak intensity. Since the molecular formula assignments in this study only included C, H, O and N, thus excluding S containing formulas, these peaks cannot be $O_3S_1$ peaks. However, we cannot exclude that these peaks were other contaminants, and as reviewer pointed out, these peaks inaccurately dominated in the former FT-ICR MS data version. After careful consideration, we decided to remove the unusually high relative peak intensities (> 0.10) from the FT-ICR MS analysis. We added an explanation of the peak removal to Methods (page 6 line 181).

**At this point it isn't clear where your study system is situated compared to the systems you say are highly and less studied; the difference between "boreal catchments" (line 44) and "northern high-latitude streams" (line 46) is not clear—boreal catchments certainly are one type of northern high-latitude stream. In line 71 you say "subarctic rivers," which might be a good description to use in line 46 to contrast with boreal catchments. Further, right now you cite five sources for boreal catchments and five for high-latitude streams, completely contradicting the point you are making in the text that one is more studied than the other. Perhaps you don't need to set it up as an either/or scenario, but simply explain why it is good to study subarctic rivers.**

Author response: We agree with the reviewer that the former sentence was misleading, as the term "high-latitude streams" can refer to both boreal and subarctic streams. We have now edited this sentence and used the term "subarctic" (page 2 line 44).

**Section 2.6: I have a few concerns about the FT-ICR MS methods you describe, and it would be helpful if you could clarify these points in this methods section. Why was only m/z range 150-500 analyzed? Normally DOM masses extend well into m/z 800-1000. Please add a short (even half a line long) explanation. Further, it sounds as if formulae were assigned one by one rather than in homologous series (the standard and far more powerful way of assigning formulae). If that is true, please justify it, or add some acknowledgement of how this may impact the results. If anyone is undertaking ultrahigh resolution mass spectrometry, they should work with the best software possible to ensure that their assignments are correct, and homologous series assignments are far more accurate than single formulae assignments. If, however, for some reason that is not what was done in this study, it at least needs to be acknowledged explicity in the paper with a small explanation.**

Author response: We thank the reviewer for these important comments concerning FT-ICR MS analysis. We acknowledge that several studies in northern freshwater ecosystems have used wider m/z range in their analyses (e.g., up to m/z 800-1000). On the other hand, previous studies have found that intensity maxima of natural organic matter (NOM) typically occurs in the range of m/z 350–500 (Reemtsma 2009), which was the case also in our study (see Fig. 1 presenting an example of the sample spectra in our FT-ICR MS analysis). Based on these factors, we decided to use the mass range of m/z 150-500 in this analysis. We have added an explanation of the m/z range selection to the revised manuscript (page 6 lines 175-177).

[Figure]

**Figure 1. An example of the mass spectra in a) blanks and b) river water samples in this study.**

We also thank the reviewer for the comment concerning the use of different molecular formula assignment strategies. We agree with the reviewer that molecular assignment using homologous series is certainly a powerful method and widely used in recent studies. However, one-by-one molecular formula assignment is still a commonly used method, and no major problems have been reported. In addition, since we aimed to evaluate the molecular composition of natural water in this study, we wanted to assign not only the molecules that could form a series, but also other molecules under the same conditions. Therefore, we used the one-by-one method in this study.

While we acknowledge the weaknesses of this approach (e.g., Koch et al. 2007), we have conducted other measures that significantly decrease the possibility of assignment errors in our FT-ICR MS data analysis. For example, since high errors are associated with the assignments containing S and P (Mostovaya et al. 2017), these formulas were excluded from further analysis. In this study, the analyses were restricted to molecular formulas with C, H, O, and max. 2 N, which can be considered to significantly decrease the possibility of false assignments (Koch et al. 2007). Furthermore, considering that the number of possible molecular formulas as well as the number of assignment errors increase with increasing m/z (Stenson et al. 2003; Sleigther et al. 2008), the use of m/z range below 500 Da also decreases the possibility of assignment errors.

**Section 2.7: I believe this is not the normal way of counting bacterial abundance, and flow cytometry would have been far more accurate. Please add some explanation of why this method was chosen to assess abundance.**

Author response:

We thank the reviewer for this comment concerning different bacterial quantification approaches and their accuracy. In previous studies, methods using fluorescent dyes to stain the bacterial cells and qPCR have been compared, and e.g. Chen and Li (2005) conclude that concentrations measured by real-time qPCR were highly associated with the total number concentrations measured by epifluorescence microscopy (EFM) and flow cytometry (FCM).

Zhang et al. (2017) suggest that the differences in FCM and qPCR observed in some studies could be due to the combination of the inherent biases of all experimental steps, such as the efficiency of cell/DNA extraction, PCR inhibition due to the amplification efficiency and primer specificity, or underestimated counts of bacterial cells with FCM. We acknowledge the high variability and other weaknesses that qPCR might include. However, in this study, the bacterial abundance was quantified to use as an estimation to compare the relative differences between the studied rivers, and therefore, we consider the accuracy of this method sufficient for these purposes. In addition, considering the remote location of the sampling area, the use of 16S qPCR with frozen glass microfiber filters was a practical approach to quantify the bacterial abundance.

**Lines 155-157: Please include whether the CO2/DOC ratio is using the DOC concentration at the beginning or end of the measurement period.**

Author response: We thank the reviewer for pointing this out. The $CO_2$/DOC ratio (i.e., the cumulative $CO_2$ production per DOC) was calculated based on the $CO_2$ production rates ($\mu mol\ l^{-1}\ d^{-1}$) measured during the four 24 h gas samplings divided by the DOC concentration in each bottle at each timepoint. We have clarified this in the text (page 5 line 156).

**Lines 244-251: Please specify whether you are talking about the mean or mean weighted average m/z, O/C, and H/C in this section. In Table 1, you say mean, but it's not clear how that is calculated. Did you add up all the m/zs for each sample and divide by the number of samples? That's what mean implies. If you mean mean weighted average, as in the mean m/z weighted by the relative abundances, please specify that. Same for O/C and H/C. At this point it's unclear what this metric is.**

Author response: Thank you for this comment. Indeed, the values presented in Table 1 are the mean values of m/z , O/C, H/C, and $AI_{mod}$, i.e. the sum of the values divided by the number of samples (n = 6 in the brown-water river and n = 5 in the clearwater river).

**Lines 252-253: Do you mean the percent of formulae based on number of formulae, or based on percent relative abundance? They can mean very different things. Please clarify.**

Author response: We thank the reviewer for this comment. The percentages of molecular formulas are based on the number of assigned molecular formulas. This has been clarified to manuscript (page 12 line 262). We also included an explanation of this to the caption of Figure 4.

**Lines 254-257: Relative peak intensities by definition add up to 100% for a single spectrum, so I do not understand how they could be higher in one river type than another. Do you mean the average relative peak intensity?**

Author response: We agree with the reviewer that the comparison of the average relative intensities was not clearly presented. After recalculating the statistical analyses due to the removal of unusually high peaks, we decided to exclude the peak intensity comparison from the chapter 3.2 and focus on the comparison of the parameters presented in Table 1.

**Lines 257-259: same as above—do you mean weighted average Almod? You need to specify. There is not just one Almod value for each sample, which is what this sounds like.**

Author response: Thank you for this comment. We have used the mean values of $AI_{mod}$ for each sample. We have now explained this in the revised manuscript (page 12 line 264). In addition, we have edited the whole paragraph concerning the significant differences in $AI_{mod}$ between the rivers due to the recalculation of statistical analyses.

**Lines 264-779 (and Figure 4): The percentages that are in this section appear to be percent of total molecular formulae in each compound class. Normally in this field folks refer to percent total relative abundances. These two concepts convey different things, and which you want to use depends on what you're looking at (number of specific/rare**

**formulae versus the contribution that compound class makes to the overall DOM signal). You might want to think about switching to percent relative abundance if you want other papers to be able to cite this for comparison, since most folks work in %relative abundance. A caveat to this is in my next comment—the massive signal peaks you are assigning as aliphatics that are probably surfactant contaminants (O3S1 class, etc) found in most FT-ICR MS spectra that use SPE. These peaks should be removed before calculating %relative abundance.**

Author response: We thank the reviewer for this comment. In this section, we present the percentages of different compound groups based on the number of molecular formulas, similarly to e.g. O'Donnel et al. (2016). We agree with the reviewer that % relative abundance is a typical way of presenting these results in this field. The solution to present the relative differences of compound groups in a current way (% of different compound groups) was based on the suggestions and critiques that %RA approach has previously received from the reviewers. Recently, there has been discussion about the use and misuse of peak intensities in organic matter chemistry (e.g., Kew et al. 2022). Therefore, while we acknowledge the benefits of using %RA (i.e., the comparability to other studies), we think that the current way of presenting the data in Figure 4 is reliable and gives a good overview of the quality and composition of DOM in these rivers.

**Lines 346-347: Unlike what you are stating in this line, Figure 5 looks to me like the two rivers do not have significant differences in bacterial abundance (just very different ranges). Could you add in some statistic earlier to show they are significantly different, if they are?**

Author response: We thank the reviewer for pointing this out. We have edited this statement and removed the part of the sentence concerning the comparison of bacterial abundances due to the lack of significance (page 17 line 354).

**Figure 3: First, the font is too small to read, and the dots of color indicating compound category in the legend are too small to see. Please make the figure and legend legible. Of more concern are the high relative abundance peaks at high H/C low O/C ratio (top left**

**corner)—these look a lot like the surfactant contaminants that are common in FT-ICR MS analysis, and belong to the O3S1 or O4S1 classes of homologous series. Please see if these peaks are O3S1, and remove the contamination series before analysis.**

Author response: We thank the reviewer for suggesting these improvements to Figure 3. In the revised figure, we have increased the font size, as well as the size of the dots indicating compound classes in the legend.

As discussed above, we checked all the peaks with unusually high relative peak intensity, and these peaks were not known contaminant peaks from $O_3S_1$ or $O_4S_1$ classes. However, since we cannot completely exclude some other possible contamination from sample preparation or other steps of the FT-ICR MS analysis, and the high peaks can distort the results and their interpretation, we decided to remove the peaks with unusually high relative peak intensity (> 0.10). Similar removal of unusually high peaks has been done also in previous studies, e.g., Mostovaya et al. (2017). We have updated all the figures, tables and statistics excluding these peaks from further analysis.

**References**

Chen, P. S., & Li, C. S. (2005). Real-time quantitative PCR with gene probe, fluorochrome and flow cytometry for microorganism analysis. Journal of Environmental Monitoring, 7(3), 257-262.

Kew, W., Myers-Pigg, A., Chang, C., Colby, S., Eder, J., Tfaily, M., et al. (2022). Reviews and syntheses: Use and misuse of peak intensities from high resolution mass spectrometry in organic matter studies: opportunities for robust usage. EGUsphere, 1-26.

Koch, B. P., Dittmar, T., Witt, M., & Kattner, G. (2007). Fundamentals of molecular formula assignment to ultrahigh resolution mass data of natural organic matter. Analytical Chemistry, 79(4), 1758-1763.

Mostovaya, A., Hawkes, J. A., Dittmar, T., & Tranvik, L. J. (2017). Molecular determinants of dissolved organic matter reactivity in lake water. Frontiers in Earth Science, 5, 106.

O'Donnell, J. A., Aiken, G. R., Butler, K. D., Guillemette, F., Podgorski, D. C., & Spencer, R. G. (2016). DOM composition and transformation in boreal forest soils: The effects of temperature

and organic-horizon decomposition state. Journal of Geophysical Research: Biogeosciences, 121(10), 2727-2744.

Reemtsma, T. 2009. Determination of molecular formulas of natural organic matter molecules by (ultra-) high-resolution mass spectrometry: status and needs. Journal of chromatography A, 1216, 3687-3701.

Sleighter, R. L., McKee, G. A., Liu, Z., & Hatcher, P. G. (2008). Naturally present fatty acids as internal calibrants for Fourier transform mass spectra of dissolved organic matter. Limnology and Oceanography: Methods, 6(6), 246-253.

Stenson, A. C., Marshall, A. G., & Cooper, W. T. (2003). Exact masses and chemical formulas of individual Suwannee River fulvic acids from ultrahigh resolution electrospray ionization Fourier transform ion cyclotron resonance mass spectra. Analytical chemistry, 75(6), 1275-1284.

Zhang, Z., Qu, Y., Li, S., Feng, K., Wang, S., Cai, W., et al. (2017). Soil bacterial quantification approaches coupling with relative abundances reflecting the changes of taxa. Scientific reports, 7, 1-11.

---

## Author Comment (AC2)

Manuscript No.: bg-2022-225

**Response to RC2**

Dear Reviewer #2,

We thank the reviewer for providing valuable comments for this manuscript. We have carefully considered the comments and incorporated most of the changes made by the reviewer. Please see below for a point-by-point response to the reviewer's comments and concerns.

**Reviewer's comments to the authors:**

**Reviewer #2**

**The manuscript "Dissolved organic matter composition regulates microbial degradation and carbon dioxide production in pristine subarctic rivers" by Saarela et al. provides a nice comparison between DOM and CO2 production in the a clear water vs brown water systems in the high latitude watershed. Studies linking DOM and CO2 has become very crucial in recent years with increasing amount of greenhouse gas emission from inland waters, and this manuscript provides valuable findings on the topic. The manuscript includes a number of advanced techniques including CO2 measurement, FT-ICR-MS, and qPCR, which combined with appropriate statistical analysis seems adequate to support the major findings. The manuscript will be valuable addition in the field of aquatic biogeochemistry and will benefit the readers of Biogeosciences. I have a few suggestions for the authors to consider before the final publication of the manuscript.**

Author response: We thank the reviewer for the suggestions to improve this manuscript.

**1. In lines 108-109, the author mention about adding an inoculum from the surface sediment. Since river water samples are usually added as inoculum for incubation experiments, please provide a brief explanation for adding inoculum from the sediment.**

Author response: We thank the reviewer for this comment concerning the use of an inoculum. Surface sediment was collected for the use as an inoculum because we considered the sediment to be more suitable to accelerate the microbial activity in the incubation experiment.

**2. Also, a previous meta-analysis study on BDOC measurement method (Vonk et al. 2015) reported no significant difference between the BDOC measured with or without inoculum when a 0.7 μm filter like GFF is used. The author could have avoided adding inoculum since enough microbes pass through the filter required for microbial degradation.**

Author response: We thank the reviewer for pointing out the interesting results by Vonk et al. (2015). While the use of microbial inoculum is a common way of conducting incubation experiments, this important issue should be considered in future experimental designs.

**3. The incubation experiment for measurement of CO2 is quite interesting. A little more details on the calculations methods or showing the actual data on a SI table would be helpful for the readers. Also, why did the author use a one-point calibration when at least two point is more usual for calibration.**

Author response: We thank the reviewer for pointing out these issues concerning the measurements of $CO_2$. We have clarified the details and equations of calculations concerning $CO_2$ production to Supplementary methods (see revised Supplementary material page 6). The data of the $CO_2$ production, as well as other data presented in manuscript figures and tables, will be published and made freely available for the readers via public data repository according to journal's submission guidelines.

In this study, we conducted a one-point calibration with a standard gas (2.02 ppm of $CH_4$, 398 ppm of $CO_2$ and 0.836 ppm of $N_2O$) due to relatively low concentrations of $CO_2$. The linearity of the calibration with a larger amount of standard gas concentrations have been regularly verified for this GC-MS analyzer.

**4. Please add a relevant reference for the compound classes assignment criteria (line 181-184).**

Author response: We thank the reviewer for this comment. A reference for the compound class assignment criteria has been added (page 6 line 188).

**5. In the results, I see a lot of statistical analysis results (i.e., p values); however, I miss seeing the actual values of the major parameters particularly in comparison between seasons or water types. Including some actual values for DOC, CO2 etc. in the result section would improve the readability of the manuscript, whereas adding the statistical results in the figure would also be helpful.**

Author response: We thank the reviewer for this important suggestion to improve the readability of this manuscript. The average values of the statistically significant major parameters have now been added to the manuscript section "3.1 Microbial degradability of DOM". In addition, statistically significant differences between the rivers (i.e., p values indicated by asterisks) have been added to Figures 1 and 2.

**6. The discussion is well supported with references, but you may add the following reference to support the findings on lability of molecular composition (example: line 344-349). Begum et al. 2022 (https://doi.org/10.1016/j.watres.2022.119362).**

Author response: We thank the reviewer for suggesting this interesting review study by Begum et al. (2022) as a reference to support our findings. This paper is highly relevant to our study, and we have now included it as a reference in our manuscript.